# Associations between Levels of Urinary Oxidative Stress of 8-OHdG and Risk of Atopic Diseases in Children

**DOI:** 10.3390/ijerph17218207

**Published:** 2020-11-06

**Authors:** Pang-Yen Chen, Chien-Wei Chen, Yu-Jang Su, Wen-Han Chang, Wei-Fong Kao, Chen-Chang Yang, I-Jen Wang

**Affiliations:** 1Department of Emergency Medicine, Mackay Memorial Hospital, Taipei 10449, Taiwan; pongyen2000@gmail.com (P.-Y.C.); pioneermd@gmail.com (Y.-J.S.); branden888@gmail.com (W.-H.C.); 2Institute of Environmental and Occupational Health Sciences, School of Medicine, National Yang-Ming University, Taipei 112, Taiwan; ccyang@vghtpe.gov.tw; 3Institute of Public Health, School of Medicine, National Yang-Ming University, Taipei 112, Taiwan; 4Department of Nursing, Yuanpei University of Medical Technology, Hsinchu 300, Taiwan; 5Department of Diagnostic Radiology, Chang Gung Memorial Hospital Chiayi Branch, Chiayi 613, Taiwan; chienwei33@gmail.com; 6College of Medicine, Chang Gung University, Taoyuan 333, Taiwan; 7Institute of Medicine, Chung Shan Medical University, Taichung 40201, Taiwan; 8Department of Medicine, Mackay Medical College, New Taipei City 252, Taiwan; 9Mackay Junior College of Medicine, Nursing, and Management, Taipei 11260, Taiwan; 10Graduate Institute of Injury Prevention and Control, Taipei Medical University, Taipei 110, Taiwan; 11Institute of Mechatronic Engineering, National Taipei University of Technology, Taipei 106, Taiwan; 12Department of Emergency Medicine, School of Medicine, Taipei Medical University, Taipei 110, Taiwan; wfkao@tmu.edu.tw; 13Department of Emergency and Critical Care Medicine, Taipei Medical University Hospital, Taipei 110301, Taiwan; 14Department of Internal Medicine, Taipei Veterans General Hospital, Taipei 112, Taiwan; 15Department of Pediatrics, Taipei Hospital, Ministry of Health and Welfare, Taipei 242, Taiwan; 16College of Public Health, China Medical University, Taichung 404, Taiwan; 17National Taiwan University Hospital, National Taiwan University, Taipei 100, Taiwan

**Keywords:** asthma, atopic dermatitis, atopic diseases, oxidative stress, urinary 8-OhdG

## Abstract

The oxidative stress biomarker of urinary 8-hydroxy-2′-deoxyguanosine (8-OHdG) was reported to be changed in patients with allergic diseases. Measurement of urinary oxidative products is noninvasive. However, correlations between the severity levels of atopic diseases and oxidative stress remain unclear. This study aimed to investigate the association among urinary 8-OHdG, atopic dermatitis (AD), and the phenotypes of atopic diseases in children. In a nested case-control study, participants of kindergarten children were enrolled from the Childhood Environment and Allergic Diseases Study (CEAS). Urinary analyses and urinary 8-OHdG were performed on samples from 200 children with AD as cases and 200 age- and sex-matched controls. Our study presents the following main findings: (1) The urinary 8-OHdG levels were significantly higher in cases than controls. Higher urinary 8-OHdG levels were associated with the risk of AD in a dose-response-manner; (2) Children’s AD history was associated with higher risks of asthma, allergic rhinitis, and night pruritus; (3) For children with AD, urinary 8-OHdG levels of >75th percentile were associated with higher risk of asthma, compared with the reference group of 0–25th percentiles. In summary, this study provides better understanding of the underlying mechanisms of AD and urinary 8-OHdG by analyzing a large-scale sample survey in Taiwan.

## 1. Introduction

Atopic diseases frequently present with the onset of atopic dermatitis (AD), asthma, pruritus, and allergic rhinitis, with significant costs to patients and their families and adverse effects on patients’ quality of life. AD is a chronic and relapsing inflammatory skin disease, which is associated with increased serum Immunoglobulin E (IgE) levels and tissue eosinophilia [1]. AD frequently predates the onset of the aforementioned conditions. The disease course is complex and often difficult to manage. To date, the current mainstay treatments include topical moisturizers, topical corticosteroids, topical calcineurin inhibitors, phototherapy, and systemic immunotherapies [2]. AD has been classified into three phases: infantile, childhood, and adult; each phase can be characterized with various physical findings [3]. In most countries, AD now affects more than 10% of children at some point during childhood [4]. AD is a chronic inflammatory disease of the skin, which often precedes asthma and other atopic diseases including asthma and allergic rhinitis, i.e., the so-called “atopic march” [5]. Currently, the values of serum IgE and eosinophils in the peripheral blood are highly sensitive in the detection of AD [1]. However, this approach has low specificity, and blood draws in children are difficult. A noninvasive way to detect the risk of AD for children is necessary.

Prior studies have shown that individuals with lowered antioxidant capacity were at an increased risk of AD [6]. Inflammatory cell activation and dysregulated cytokine production appear to play critical roles in the pathogenesis of AD [7]. In this setting, alterations in the flux of free radicals and oxidative stress can have profound influences on the pathophysiology of AD [7,8,9]. Oxidative stress, with the formation of reactive oxygen species (ROS), is the key element of inflammation. The oxidative stress biomarker of urinary 8-hydroxy-2′-deoxyguanosine (8-OHdG) was reported to be changed in patients with allergic diseases [10]. Measurements of urinary oxidative products are noninvasive. However, correlations between the severity levels of atopic diseases and oxidative stress remain unclear, and studies on Asian populations are lacking. As moderate-to-severe AD can lead to poor quality of life, the development of targeted immunomodulators should remain a priority. So far, the treatment of moderate-to-severe AD relies on potent corticosteroids and systemic immunosuppressants [11]. Children on systemic treatment require blood draws to manage the disease. A better understanding of the underlying mechanisms of AD is therefore crucial for designing new and more effective treatments.

Firstly, this study aimed to investigate the association among urinary 8-OHdG levels, AD, and the phenotypes of atopic diseases in children. Secondly, we aimed to study the associations between urinary 8-OHdG levels and the phenotypes of atopic diseases in children with AD. In this study, we hypothesized that impaired homeostasis of oxygen radicals may be involved in the progression of the inflammatory process in childhood AD. Further understanding and confirmation of their relationship could help early management of the disease in the future.

## 2. Materials and Methods

### 2.1. Study Design and Study Population

In a nested case-control study, a total of 453 participants comprising kindergarten children aged 3–6 years residing in communities of Northern Taiwan from the Childhood Environment and Allergic Diseases Study (CEAS) were enrolled in 2011 [12]. Overall, 200 children were identified as AD cases. Controls were then 1 to 1 matched to the same diagnosed time of AD cases. Participants who met the following criteria were excluded: inability to answer questions in Chinese, congenital diseases, acute febrile illness, upper respiratory infection (stuffy or runny nose, sore throat, cough, etc.), and childhood diabetes mellitus, other inflammatory skin conditions, and cancer. Table 1 provided the baseline characteristics of the study participants.

This study was approved by the Institutional Review Board of Taipei Hospital, Ministry of Health and Welfare, Taiwan, R.O.C. (IRB number: TH-IRB-09-04), we obtained informed consents from all participants’ parents.

### 2.2. Definitions of Cases, Controls, and Types of Atopic Diseases

A diagnosis of AD, asthma, and allergic rhinitis was made by expert pediatric allergists using a standardized technique comprising history taking and clinical examinations. The standard International Study of Asthma and Allergies in Childhood (ISAAC) questionnaires were answered by the children’s parents [12]. The cases of AD were confirmed according to the diagnostic criteria developed by Hanifin and Rajka [13]. Eosinophil count, total Immunoglobulin E (IgE), allergen specific IgE, SCORAD (SCORing Atopic Dermatitis) index were applied in the context of reported atopic dermatitis. Children without AD were selected as normal controls.

Allergic rhinitis was confirmed using the Allergic Rhinitis and its Impact on Asthma (ARIA) guidelines [14]. Asthma was diagnosed by pediatric allergists based on at least one of the three criteria: (1) Children were with recurrence of at least two symptoms of cough, wheeze, and shortness of breath, but without having a cold within the last 12 months; (2) Children were receiving ongoing treatment of asthma; (3) Children were response to treatment with β2-agonists or inhaled corticosteroids [15]. Night pruritus was identified as sleep disturbance due to skin irritation collecting via the ISAAC questionnaires: “In the last 12 months, has your child suffered from night pruritus? How many times per week has your child been kept awake at night by the symptom of itchy rash?” [16].

### 2.3. Data Collection and Laboratory Methods for Urine Samples

A standardized questionnaire was used to collect information from the participants’ parents regarding the history and symptoms of atopic diseases in children, environmental exposures (e.g., furry pets/carpets at home, exposures of environmental) tobacco smoke, children’s age, sex, child birth history, body mass index (BMI), preterm birth <37 weeks. The mothers’ information of nationality, parental age, parental history of atopy, education levels (junior high school and below/senior high school and above), family income (<USD $20,000/USD $20,000–33,300/≥USD $33,300), and duration of breast feeding <6 months/≥6 months) were also collected [12]. The z-scores (standard deviation scores) and percentages of BMI were measured based on the child’s BMI, age and sex, and can be checked online: [17].

We collected early-morning void urine samples from all participants at enrollment in 2011. Urinary 8-OHdG was measured after the occurrence of AD, and the diagnoses of AD and other phenotypes of atopic diseases was performed at enrollment. The samples were stored at −20 °C until analysis. The concentrations of 8-OHdG were measured using a competitive enzyme-linked immunosorbent assay (ELISA) kit (8-OHdG Check, Japan Institute for the Control of Aging, Fukuroi, Shizuoka, Japan) as described previously [18]. Briefly, 10µL of urine was diluted 20-fold before analysis. Urine samples were measured in duplicate in a 96-well microplate format and a standard protocol. The inter- and intra- assay coefficients of variation were less than 10%. Total urinary 8-OHdG concentrations were measured by multiplying the concentration with the total volume of 12-h urine. The sample urinary creatinine concentrations were normalized to assess changes and variations in urine concentration.

### 2.4. Statistical Analysis

Data were expressed as mean ± standard deviation for normally distributed, continuous variables and as proportions for categorical variables. Continuous variables were analyzed using a two-tailed t-test. Nonparametric parameters were tested using the Mann-Whitney U test. Discrete variables were compared using a Chi-square test. The concentrations of 8-OHdG were divided into quartiles for analysis, with the lowest quartile used as the reference category.

A conditional logistic regression was performed to evaluate the associations of 8-OHdG levels and the risks of atopic diseases in both univariate and multivariate models. For Table 2 and Table 3, the risk factors adjusted in the multivariate regression model (Table 3-Model 1) were selected from the univariate model (Table 2-Model 0) predicting any one of atopic diseases (atopic dermatitis, asthma, allergic rhinitis, night pruritus) based on the risk factor with a significance level of *p*-value ≤ 0.2 plus age and gender for all cases and controls (*n* = 400). Model 1 (multivariate regression model) was adjusted by age, gender, daycare before 1-year-old, furry pets at home, tobacco smoke exposure during pregnancy, and parental history of atopy (due to 8-OHdG level was correlated to AD result, the status of AD cannot be simultaneously considered as one of the confounders in the same model with 8-OHdG level. As a result, the levels of continuous urinary 8-OHdG and the status of atopic dermatitis were adjusted in Model 1 independently). For Table 4 and Table 5, odds ratios (ORs) were adjusted for potential confounders of children’s age, children’s gender, daycare before 1-year-old, preterm birth <37 weeks, family income (<USD $20,000/USD $20,000–33,300/≥USD $33,300), duration of breast feeding <6 months/≥6 months), tobacco smoke exposure during pregnancy, and parental history of atopy to explore the associations between 8-OHdG quartiles and atopic phenotypes in cases (*n* = 200). Pearson’s correlation coefficient (r) was used to measure the strength of the association between the two variables. A *p*-value of <0.05 was considered as statistically significant. Statistical analysis was performed with the SAS (version 9.4.; SAS Institute Inc., Cary, NC, USA). The power of this study was 98.2%, as assessed by the online post hoc Power Calculator [19].

## 3. Results

### 3.1. Demographic Characteristics

This study involved 200 AD cases and 200 controls. Children with AD were enrolled at average age of 5.55 ± 1.25 years old with 58% boys. The distributions of age, sex, premature birth (<37 weeks), birth body weight, BMI, maternal history of atopy, and environmental exposures of duration of breast feeding, vitamins C or E intakes, furry pets at home, carpets at home, fungus on house wall, exposures of environmental tobacco smoke, and family income were comparable between cases and controls. For cases, 68.5% children were with urinary 8-OHdG levels >21.4 (ng/mg Cr) nanograms per milligram creatinine, whereas only 48.5% controls were with urinary 8-OHdG levels >21.4 ng/mg Cr (*p* < 0.001; Table 1).

The level of urinary 8-OHdG in AD children was significantly higher than controls (35.6 ± 27.7 vs 26.2 ± 17.6 ng/mg creatinine (Cr) respectively; *p* < 0.001; Table 1). As compared with controls, more children with AD were sent to daycare center at the age of less than 1-year-old (29.5% in cases vs 19.5% in controls, respectively; *p* = 0.02). In addition, cases were associated with higher rates of asthma, allergic rhinitis, and night pruritus then controls (all *p* < 0.001, Table 1).

### 3.2. Relations of Urinary 8-OHdG to the Risks of Atopic Diseases

Table 2 shows the associations between urinary 8-OHdG levels and the risk of AD and asthma without adjustment in cases and controls. Table 3 reports adjusted associations between urinary 8-OHdG levels and atopic diseases. The adjusted OR of AD was found to increase with higher urinary 8-OHdG levels in a dose-response-manner (OR: 1.02, 95% CI:1.01–1.03, *p* < 0.001; Table 3). When the urinary 8-OHdG level was higher than 37.61 ng/mg Cr (>75th percentiles), the adjusted OR of AD risk significantly increased to 2.99 (95% CI:1.67–5.38) as compared with reference group of urinary 8-OHdG <16.14 ng/mg Cr, 0–25th percentiles. In addition, AD was less likely to occur in male children than females (OR:0.63, 95% CI:0.41–0.97, *p* = 0.03; Table 3). On the other hand, urinary 8-OHdG level was not significantly associated with asthma risk.

Higher urinary 8-OHdG levels did not significantly increase the risks of allergic rhinitis and night pruritus (Table 3). However, parental maternal history of atopy was associated with higher risks of allergic rhinitis and night pruritus in children (OR:1.75, 95% CI:1.07–2.84, *p* = 0.03 for allergic rhinitis; OR:2.03, 95% CI:1.17–3.52, *p* = 0.01 for night pruritus). Additionally, children with a history of AD were more likely to have diagnoses of asthma, allergic rhinitis, and night pruritus (*p* < 0.001; Table 3).

### 3.3. The Associations of Urinary 8-OHdG Levels and Atopic Phenotypes in Children with AD

Table 4 exhibited the associations of urinary 8-OHdG quartiles and atopic phenotypes in children with AD. AD children with higher urinary 8-OHdG levels >42.7 ng/mg Cr, >75th percentiles) were significantly associated with asthma (adjusted OR, 95% CI:2.71, 1.14–6.43) as compared with reference group of urinary 8-OHdG <17.68 ng/mg Cr, 0–25th percentiles; *p*-value = 0.02). In addition, there was no positive association between urinary 8-OHdG levels and asthma, allergic rhinitis, or night pruritus (Table 4).

For all children, irrespective of a diagnosis of AD, higher levels of urinary 8-OHdG significantly increased with higher frequency of exposures of environmental tobacco smoke (r = 0.14, *p* = 0.04). For all children with AD, the frequency of night pruritus was positively correlated with 8-OHdG levels (r = 0.17, *p* = 0.01), although it was failed to reach statistical significance after adjusting for multivariate confounders. Additionally, no gender difference among various urinary 8-OHdG levels was found in children with AD (r = 0.12, *p* = 0.09). For control children without AD, urinary 8-OHdG levels were not associated with any atopic phenotypes (refer to Table 5).

## 4. Discussion

Our study presents the following main findings: (1) The urinary 8-OHdG levels were significantly higher in children with AD than controls. Higher urinary 8-OHdG levels were associated with the risk of AD in a dose-response-manner; (2) Children’s AD history was associated with higher risks of asthma, allergic rhinitis, and night pruritus; (3) For children with AD, urinary 8-OHdG levels of >75th percentiles were associated with higher risk of asthma, compared with reference group of 0–25th percentiles.

AD is a chronic skin inflammatory which is characterized by increased serum IgE levels, tissue eosinophilia, intense infiltration of lymphocytes, and monocytes [18,20]. Inflammatory processes lead to an imbalance between oxidant and antioxidant components. A review observed that an imbalance between oxidant and antioxidant components may generate an excessive amount of ROS, reactive nitrogen species (RNS), and free radicals (so-called “oxidative stress”), and result in allergic and irritant dermatitis [21]. Oxidative stress drives the production of oxidation products which can denature proteins, alter apoptosis of cells, and influence the release of proinflammatory mediators, which may be critical for the induction of some inflammatory skin diseases [22,23]. Urinary 8-OHdG is an oxidized nucleoside and a marker of oxidative deoxyribonucleic acid (DNA) damage, which has been used as an accepted biomarker of oxidative stress [24]. Excessive exposures to ROS and RNS are the hallmarks of oxidative stress, and lead to the damage of proteins, lipids, DNA and other macromolecules. Once DNA is damaged and repaired, urinary 8-OHdG will be excreted in the urine [10,24]. Therefore, the urinary level of 8-OHdG can reflect the extent of oxidative DNA damage [6,10,18,21,25,26]. As a result, increased concentrations of urinary 8-OHdG as the index of the imbalance between oxidant and antioxidant components also reflect the development of inflammatory skin diseases. Prior studies have reported that increased urinary 8-OHdG concentrations were significantly higher and nitrite/nitrate levels were significantly lower in AD children than in controls, suggesting that increased oxidative stress and impaired oxygen/nitrogen radicals are involved in the progress of childhood AD [10,18,21]. Better understanding the underlying mechanisms of AD is important. Currently, the values of serum IgE and eosinophils in the peripheral blood are highly sensitive in the detection of AD. Children under systemic treatment requires blood draws. Considering the clinical application of urinary 8-OHdG in children with AD who need to receive blood draws during follow-up, we can simply examine the concentrations of 8-OHdG via urinary samples. Regarding clinical application in children with AD, a urinary concentration of 8-OHdG could be considered one of the surrogate indexes of a non-invasive method for detecting AD activity.

Prior studies have presumed that atopic diseases including AD, asthma, and allergic rhinitis are mediated by oxidative stress. [7,27,28]. Approximately 50% of AD children will develop asthma, particularly those with severe AD, and around 65% will develop allergic rhinitis [3]. One prior study revealed the rate of AD was significantly higher in teenagers with asthma than those without it (risk ratio [RR], 95% CI:4.5, 3.1–6.5) [29]. Prior studies have suggested that respiratory allergy and AD have a common systemic link [30,31]. The disease severity of asthmatic children with AD was greater than that of asthmatic children without AD [32]. Some cytokines, such as IL-6, are correlated with night pruritus and sleeping quality of AD patients [33]. A mouse animal model proved the correlation of oxidative stress and mice scratching behavior due to itching [34]. The results of previous studies supported the hypothesis that the mechanisms of pruritus were associated with allergic reactions and inflammations. Patients with AD had reduced thresholds for irritants, and significantly greater irritant skin responses than healthy, nonatopic controls [35]. Increased levels of oxidative stress occur not only a result of inflammation, but also from environmental exposure to air pollution and cigarette smoke. In this study, children’s AD history was associated with higher risks of asthma, allergic rhinitis and night pruritus, indicating that the “atopic march” is characterized by the progression of atopic dermatitis to asthma and allergic rhinitis. The putative mechanism could be that chronic skin inflammation, as an imbalance between oxidant and antioxidant components, can result in primary sensitization, then breaking the sensitization in the airways through the epidermal barrier [3,36]. Recent animal studies using asthmatic mouse models reported that ovalbumin/ozone-induced asthma was associated with elevated levels of 8-OHdG [37,38,39]. By contrast, Franken et al. reported that 8-OHdG was not associated with doctor-diagnosed asthma in adolescents aged 14–15 years [40]. In our study, AD children with higher urinary 8-OHdG levels were associated with higher risk of asthma with an adjusted OR of 2.71 in children aged 3–6 years.

Our study has some limitations. Firstly, it used spot urinary 8-OHdG concentrations as a proxy for oxidative stress exposure, instead of direct monitoring. Secondly, the urinary 8-OHdG concentrations were examined just once. Our results could not determine the serial changes of urinary 8-OHdG concentrations for AD patients. Third, there were potential day-to-day variations of urinary 8-OHdG concentrations, although the excretion of 8OHdG in urine was rapid, i.e., within at least 24 h [41,42], and half-life was short (i.e., 6–35 h) [43]. Nevertheless, the reproducibility of the methods and the stability of urinary 8-OHdG were acceptable when consecutive series of determination were compared [44]. Fourth, we did not provide the degree of AD and treatment in children; the relationship between the degree of AD and the degree of 8-OHdG needs further study in order to be confirmed. Moreover, there might be other confounding factors, such as environmental tobacco smoke, which might cause increases in oxidative stress [45]. However, we controlled several potential confounding factors in this study (as shown in Table 2, Table 3 and Table 4).

To the best of our knowledge, this is the first study representing real-world epidemiologic data in Taiwan to study the associations between the levels of urinary 8-OHdG and the onset of AD in children. In addition, as compared with previous studies, the strength of our study was enhanced because of the large sample size.

## 5. Conclusions

Our study demonstrated that higher urinary 8-OHdG levels were significantly associated with higher risks of AD and asthma in children in a dose-response relationship by a large-scale sample survey in Taiwan. Regarding the clinical application in children with AD, the urinary concentration of 8-OHdG could be considered one of the surrogate indexes of noninvasive methods for detecting AD activity. This study’s essential contribution is better to understand the underlying mechanisms of AD and urinary 8-OHdG.

Further research should address oxidative stress mechanisms in developing AD and other atopic diseases using cohort studies with large sample sizes.

## Figures and Tables

**Table 1 ijerph-17-08207-t001:** General characteristics of the participants.

Characteristics	Case	Control	*p* Value
	(*n* = 200)	(*n* = 200)	
Children			
Age	5.55 ± 1.25	5.69 ± 0.93	0.19
Male gender (%)	116 (58.0%)	101 (50.5%)	0.10
Premature birth (<37 weeks) (%)	13 (6.5%)	19 (9.5%)	0.26
Birth body weight (gm)	3199 ± 413	3104 ± 493	0.49
Body mass index (BMI)	16.2 ± 1.90	16.0 ± 1.91	0.32
Z-score of BMI	0.18 ± 1.12	0.30 ± 1.14	0.34
Percentage of BMI	55.9 ± 31.5	59.4 ± 28.8	0.28
Urinary 8-OHdG (ng/mg Cr)	35.6 ± 27.7	26.2 ± 17.6	<0.001
Asthma (%)	49 (25.4%)	4 (2.00%)	<0.001
Allergic rhinitis (%)	99 (50.8%)	8 (4.00%)	<0.001
Night pruritus (%)	71 (35.5%)	2 (1.00%)	<0.001
Mothers			
Maternal age at delivery (years)	30.0 ± 4.19	30.2 ± 4.35	0.20
Maternal education (%) *			
Junior high school and below	6 (3.0%)	7 (4.0%)	0.79
Senior high school and above	191 (97.0%)	192 (96.0%)	
Maternal Nationality (%) *			
Taiwan	188 (96.4%)	188 (94.4%)	0.36
Foreign countries	7 (3.60%)	11 (5.60%)	
Parental history of atopy (%)	63 (31.5%)	81 (40.5%)	0.26
Environment			
Duration of breast feeding (%) *			
No	35 (18.1%)	43 (22.2%)	0.61
<6 months	110 (57.0%)	105 (54.1%)	
≥6 months	48 (24.9%)	46 (23.7%)	
Daily additional supplementation of vitamin C or E intake (%)	67 (37.0%)	76 (40.4%)	0.49
Number of older siblings < 2 (%)	160 (80.0%)	170 (85.0%)	0.31
Daycare < 1-year-old (%)	59 (29.5%)	39 (19.5%)	0.02
Furry pets at home (%)	29 (14.5%)	43 (21.5%)	0.08
Carpets at home (%)	17 (8.50%)	13 (6.50%)	0.39
Fungus on house wall (%)	81 (40.5%)	73 (36.5%)	0.36
Exposures of environmental tobacco smoke (%)	112 (56.0%)	102 (51.0%)	0.24
Family income per month ($USD) (%) *			
<20,000	62 (34.1%)	64 (33.5%)	0.39
20,000~33,300	51 (28.0%)	65 (34.0%)	
≥33,300	69 (37.9%)	62 (32.5%)	

* The sum of *n* may not be equal to total *n* due to missing values; Data were presented with mean ± standard deviation or number (%).

**Table 2 ijerph-17-08207-t002:** Association between urinary 8-OHdG levels and the risk factors without adjustment (*n* = 400).

Factors	Model 0							
	Atopic Dermatitis		Asthma		Allergic Rhinitis		Night Pruritus	
	Crude OR	*p*-Value	Crude OR	*p*-Value	Crude OR	*p*-Value	Crude OR	*p*-Value
	(95% CI)		(95% CI)		(95% CI)		(95% CI)	
Urinary 8-OHdG levels (ng/mg Cr)	1.02 (1.01–1.03)	<0.001	1.00 (0.99–1.01)	0.98	1.004 (0.996–1.013)	0.33	1.003 (0.993–1.013)	0.60
Atopic dermatitis	-	-	16.7 (5.85–47.0)	<0.001	24.6 (11.5–52.7)	<0.001	54.5 (13.1–226)	<0.001
Age (years)	0.91 (0.76–1.09)	0.29	1.24 (0.96–1.60)	0.11	1.18 (0.96–1.44)	0.11	1.09 (0.87–1.37)	0.45
Gender (male)	0.71 (0.48–1.05)	0.09	0.75 (0.42–1.36)	0.34	0.78 (0.50–1.22)	0.27	0.80 (0.48–1.35)	0.41
Daycare < 1-year-old	1.56 (0.98–2.47)	0.06	1.57 (0.83–2.94)	0.16	1.31 (0.79–2.17)	0.30	1.43 (0.81–2.51)	0.21
Furry pets at home	0.62 (0.37–1.05)	0.07	0.55 (0.22–1.33)	0.18	0.74 (0.41–1.37)	0.34	0.86 (0.44–1.70)	0.66
Tobacco smoke exposure during pregnancy	4.15 (0.47–37.5)	0.20	4.48 (0.73–27.5)	0.11	1.89 (0.31–11.5)	0.49	2.93 (0.48–17.9)	0.24
Parental history of atopy	1.23 (0.83–1.84)	0.30	1.62 (0.89–2.95)	0.12	1.61 (1.01–2.56)	0.04	2.02 (1.19–3.46)	0.01
Z-score of body mass index	1.10 (0.91–1.32)	0.34	1.04 (0.79–1.37)	0.76	1.15 (0.92–1.43)	0.23	1.15 (0.89–1.48)	0.28
Preterm birth (<37 weeks)	0.65 (0.31–1.36)	0.25	0.41 (0.10–1.78)	0.24	1.03 (0.46–2.31)	0.94	1.25 (0.52–3.02)	0.62
Family income per month ($USD)		0.46		0.82		0.47		0.76
<20,000	Reference		Reference		Reference		Reference	
20,000~33,300	0.90 (0.54–1.49)	0.67	1.01 (0.49–2.07)	0.97	0.76 (0.43–1.35)	0.35	0.99 (0.52–1.90)	0.98
≥33,300	1.22 (0.75–2.00)	0.42	0.82 (0.40–1.69)	0.59	0.72 (0.42–1.26)	0.26	1.22 (0.66–2.25)	0.53
Duration of breast feeding		0.71		0.28		0.25		0.56
No	Reference		Reference		Reference		Reference	
<6 months	1.25 (0.74–2.09)	0.41	0.57 (0.28–1.14)	0.11	1.26 (0.70–2.27)	0.44	1.48 (0.72–3.04)	0.29
≥6 months	1.17 (0.64–2.13)	0.62	0.67 (0.30–1.52)	0.34	0.78 (0.39–1.59)	0.50	1.44 (0.64–3.27)	0.38

Abbreviations: CI, confidence interval; OR, odd ratio; Model 0: crude effect (without adjustment); (ng/mg Cr), nanograms per milligram of creatinine.

**Table 3 ijerph-17-08207-t003:** Association between urinary 8-OHdG levels and the risk factors in the adjusted Model 1 (*n* = 400).

Factors	Model 1							
	Atopic Dermatitis		Asthma		Allergic Rhinitis		Night Pruritus	
	Adjusted OR	*p*-Value	Adjusted OR	*p*-Value	Adjusted OR	*p*-Value	Adjusted OR	*p*-Value
	(95% CI)		(95% CI)		(95% CI)		(95% CI)	
Urinary 8-OHdG levels (ng/mg Cr) *	1.02 (1.01–1.03)	<0.001	1.00 (0.99–1.01)	0.98	1.004 (0.996-1.013)	0.33	1.003 (0.993–1.013)	0.60
Atopic dermatitis *	-	-	16.7 (5.85–47.0)	<0.001	24.6 (11.5–52.7)	<0.001	54.5 (13.1–226)	<0.001
Age (years)	0.91 (0.76–1.09)	0.29	1.24 (0.96–1.60)	0.11	1.18 (0.96–1.44)	0.11	1.09 (0.87–1.37)	0.45
Gender (male)	0.71 (0.48–1.05)	0.09	0.75 (0.42–1.36)	0.34	0.78 (0.50–1.22)	0.27	0.80 (0.48–1.35)	0.41
Daycare < 1-year-old	1.56 (0.98–2.47)	0.06	1.57 (0.83–2.94)	0.16	1.31 (0.79–2.17)	0.30	1.43 (0.81–2.51)	0.21
Furry pets at home	0.62 (0.37–1.05)	0.07	0.55 (0.22–1.33)	0.18	0.74 (0.41–1.37)	0.34	0.86 (0.44–1.70)	0.66
Tobacco smoke exposure during pregnancy	4.15 (0.47–37.5)	0.20	4.48 (0.73–27.5)	0.11	1.89 (0.31–11.5)	0.49	2.93 (0.48–17.9)	0.24
Parental history of atopy	1.23 (0.83–1.84)	0.30	1.62 (0.89–2.95)	0.12	1.61 (1.01–2.56)	0.04	2.02 (1.19–3.46)	0.01
Z-score of body mass index	1.10 (0.91–1.32)	0.34	1.04 (0.79–1.37)	0.76	1.15 (0.92–1.43)	0.23	1.15 (0.89-1.48)	0.28
Preterm birth (<37 weeks)	0.65 (0.31–1.36)	0.25	0.41 (0.10–1.78)	0.24	1.03 (0.46–2.31)	0.94	1.25 (0.52–3.02)	0.62
Family income per month ($USD)		0.46		0.82		0.47		0.76
<20,000	Reference		Reference		Reference		Reference	
20,000~33,300	0.90 (0.54–1.49)	0.67	1.01 (0.49–2.07)	0.97	0.76 (0.43–1.35)	0.35	0.99 (0.52–1.90)	0.98
≥33,300	1.22 (0.75–2.00)	0.42	0.82 (0.40–1.69)	0.59	0.72 (0.42–1.26)	0.26	1.22 (0.66–2.25)	0.53
Duration of breast feeding		0.71		0.28		0.25		0.56
No	Reference		Reference		Reference		Reference	
<6 months	1.25 (0.74–2.09)	0.41	0.57 (0.28–1.14)	0.11	1.26 (0.70–2.27)	0.44	1.48 (0.72–3.04)	0.29
≥6 months	1.17 (0.64–2.13)	0.62	0.67 (0.30–1.52)	0.34	0.78 (0.39–1.59)	0.50	1.44 (0.64–3.27)	0.38

Abbreviations: CI, confidence interval; OR, odd ratio; Model 0: crude effect (without adjustment); Model 1: adjusted by age, gender, daycare <1-year-old, furry pets at home, tobacco smoke exposure during pregnancy, and parental history of atopy. * The parameters were adjusted in Model 1 independently.

**Table 4 ijerph-17-08207-t004:** Association between urinary 8-OHdG levels and atopic phenotypes restricted to participants with atopic dermatitis (*n* = 200).

Urinary 8-OHdG Levels (ng/mg Cr)	Total	Number of Atopic Phenotypes (%)	Adjusted OR	*p*-Value
	*n* = 200		(95% CI)	
Asthma				0.02
Q1: <17.68	50	16 (32.0%)	1 (reference)	
Q2: 17.68–26.38	50	14 (28.0%)	1.17 (0.47–2.89)	
Q3: 26.38–42.7	50	8 (16.0%)	1.02 (0.40–2.60)	
Q4: >42.7	50	11 (22.0%)	2.71 (1.14–6.43)	
Allergic rhinitis				0.05
Q1: <17.68	50	32 (64.0%)	1 (reference)	
Q2: 17.68–26.38	50	28 (56.0%)	1.10 (0.49–2.49)	
Q3: 26.38–42.7	50	19 (38.0%)	1.39 (0.61–3.20)	
Q4: >42.7	50	20 (40.0%)	2.13 (0.93–4.89)	
Night pruritus				0.18
Q1: <17.68	50	21 (42.0%)	1 (reference)	
Q2: 17.68–26.38	50	18 (36.0%)	1.08 (0.46–2.54)	
Q3: 26.38–42.7	50	16 (32.0%)	1.13 (0.48–2.68)	
Q4: >42.7	50	16 (32.0%)	2.20 (0.95–5.09)	

Abbreviations: CI, confidence interval; OR, odd ratio; (ng/mg Cr), nanograms per milligram of creatinine; Q1: 0–25th percentiles; Q2: 25–50th percentiles; Q3: 50–75th percentiles; Q4: >75th percentiles; OR was adjusted by age, gender, daycare <1-year-old, preterm birth <37 weeks, family income (<USD $20,000/USD $20,000–33,300/≥USD $33,300), duration of breast feeding <6 months/≥6 months), tobacco smoke exposure during pregnancy, and parental history of atopy.

**Table 5 ijerph-17-08207-t005:** Association between urinary 8-OHdG levels and atopic phenotypes restricted to participants of controls (*n* = 200).

Urinary 8-OHdG Levels (ng/mg Cr)	Total	Number of Atopic Phenotypes (%)	Adjusted OR	*p*-Value
	*n* = 200		(95% CI)	
Asthma				0.67
Q1: <14.94	50	1 (2.0%)	1 (reference)	
Q2: 14.94–21.24	50	0 (0.0%)	0 (0-na)	
Q3: 21.24–34.18	50	2 (4.0%)	4.01 (0.16–103)	
Q4: >34.18	50	1 (2.0%)	0.79 (0.03–23.0)	
Allergic rhinitis				0.35
Q1: <14.94	50	2 (4.0%)	1 (reference)	
Q2: 14.94–21.24	50	0 (0.0%)	0 (0-na)	
Q3: 21.24–34.18	50	3 (6.0%)	3.53 (0.28–44.7)	
Q4: >34.18	50	3 (6.0%)	3.27 (0.26–42.0)	
Night pruritus				>0.99
Q1: <14.94	50	1 (2.0%)	1 (reference)	
Q2: 14.94–21.24	50	0 (0.0%)	0 (0-na)	
Q3: 21.24–34.18	50	0 (0.0%)	0 (0-na)	
Q4: >34.18	50	1 (2.0%)	1.98 (0.04–104)	

Abbreviations: CI, confidence interval; OR, odd ratio; na: not available; Q1: 0–25th percentiles; Q2: 25–50th percentiles; Q3: 50–75th percentiles; Q4: >75th percentiles; OR was adjusted by age, gender, daycare <1-year-old, preterm birth <37 weeks, family income (<USD $20,000/USD $20,000–33,300/≥USD $33,300), duration of breast feeding <6 months/≥6 months), tobacco smoke exposure during pregnancy, and parental history of atopy.

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
