# Peer review of "Associations between Levels of Urinary Oxidative Stress of 8-OHdG and Risk of Atopic Diseases in Children"

_ijerph, 2020, doi:10.3390/ijerph17218207_

Round 1
Reviewer 1 Report
Thank you for your submission
A. Suggest some English grammar changes as below:
1. Line 52 "the treatment indicator for treating AD can simply examine by using the concentration of urinary 8-OHdG" - change to "the urinary concentration of 8-OHdG can be examined"
2. Line 57 "predates the risks" - change to "predates the onset"
3. Line 59 "a complex condition" - delete "a" and "condition"
4. Line 60 "Up to date" - delete "UP"
5. Line 79 "data were lack of human studies in Asia with large sample size." - change to "studies on Asian population are lacking"
Line 93 "Participants who were" - delete "were"
Line 100 "To identify the diagnoses of atopic diseases, the history of AD, asthma, and allergic rhinitis were diagnosed by expert pediatric allergists" - change to "A diagnosis of AD, asthma and allergic rhinitis was made by expert pediatric allergists"
Line 295 "First" - change to Firstly ... similarly change second to secondly. Please do this throughout the manuscript
B. Suggest avoiding non-standard abbreviations such as "ETS"
Author Response
Open Review
Comments and Suggestions for Authors
Thank you for your submission
- Suggest some English grammar changes as below:
- Line 52 "the treatment indicator for treating AD can simply examine by using the concentration of urinary 8-OHdG" - change to "the urinary concentration of 8-OHdG can be examined"
Reply:
We revised to “the urinary concentration of 8-OHdG can be examined for managing AD treatment”.
- Line 57 "predates the risks" - change to "predates the onset"
Reply:
We have corrected it based on your suggesting.
- Line 59 "a complex condition" - delete "a" and "condition"
Reply:
We have corrected it based on your suggesting.
- Line 60 "Up to date" - delete "UP"
Reply:
We have corrected it based on your suggesting.
- Line 79 "data were lack of human studies in Asia with large sample size." - change to "studies on Asian population are lacking"
Reply:
We have corrected it based on your suggesting.
Line 93 "Participants who were" - delete "were"
Reply:
We have corrected the typo.
Line 100 "To identify the diagnoses of atopic diseases, the history of AD, asthma, and allergic rhinitis were diagnosed by expert pediatric allergists" - change to "A diagnosis of AD, asthma and allergic rhinitis was made by expert pediatric allergists"
Reply:
We have corrected it based on your suggesting.
Line 295 "First" - change to Firstly ... similarly change second to secondly. Please do this throughout the manuscript
Reply:
We have corrected it based on your suggesting.
- Suggest avoiding non-standard abbreviations such as "ETS"
Reply:
We have corrected all “ETS” to “environmental tobacco smoke”.
Reviewer 2 Report
Review: ijerph-938757
The Associations between the Levels of Urinary Oxidative Stress of 8-OHdG and the Risks of Atopic Diseases in Children
Comments and Suggestions for Authors
The paper by Pang-Yen Chen1et al. examines the role of 8-OHdG as a non-invasive marker of atopic diseases. The research questions being asked are important and experiments are pretty good executed however the originality / novelty seems average. Although the analysis of data is extensive, there are several drawbacks in the study design and presentation of the results. Statistical analysis also raises doubts.
Major Concerns and questions:
As stated above, it is not clear how some of the statistical analyses has been done.
- First: how the dominant explanatory variables for the model 1 of adjustment has been chosen. The selection should be supported with principal component analysis or other method.
Second: In several places the Tabaco smoke exposure (ETS) has been raised but it seems to be not strong association looking at case vs control values and especially that correlation coefficient is really weak r=0.14 and p=0.04.
It has been reported that 8-OHdG and 8-OHGua is much more often noted in fluids of smokers but not in their offspring. Those results also suggested that 8-OH-Gua, rather than 8-OH-dG, may be a more general marker for oxidative damage.
Much stronger influence on atopic diseases prevalence seems to have positive history of atopy. Also on this topic, it would be more appropriate to provide the result of both parents history of atopy, not only the mothers.
Third: In the comparison of some quantitative data when only case vs control has been compared the T-test should have been used, or when distribution is other than normal or variance is unequal, nonparametric Welch or Mann-Whitney tests should be applied not always Chi-square like for categorical variables.
- Survey data and clinicians anamnesis should be also supported with some immunological factors like TEWL or declared EASI scale results in the context of reported atopic dermatitis. This would make it more credible to qualify participants to particular groups which are also not entirely clear how numerous they are.
It would be also good to check another
It is unclear how the number of AD people diagnosed as simultaneously suffering from asthma, allergic rhinitis night pruritus (n=49, 99, 71 respectively- Table .1) increased to 69, 115, 97 respectively (Table 5-Number of atopic phenotypes (%)).
- The BMI scale used in the research should be replaced with the BMI z-score coefficient recommended by WHO as more appropriate for children.
- What was tested with the parameter described as the vitamin C and E intake. Was it additional supplementation with or preparations or dietary intake?
In this context it would be interesting to check how the diet (that in Asian countries is reach in antioxidants) influence the usefulness of this marker.
- It would be also good to make it clear if the control group can be classified as a healthy because there should be no atopic cases then. That way of involving of control participants seems more clear.
- Were there isolated associations of marker level in relation to the age of the patients conducted or only according to the quartiles?
In summary, the statistical reanalyzing with better explanation of obtained data seems the most important aspect that needs to be worked on before the manuscript can be published.
Other Concerns
Introduction
Be precise that the abbreviation AD refers to atopic dermatitis and not to atopic diseases. It seems sometimes not followed by authors.
Line 65. Please report more up-to-date data and dedicated on the region covered by the study.
Materials and Methods :
Line 123: the volume of urine and amount of consumed water during the day before should be taken into consideration when the final dilution of the urine was applied.
Line 124: not “0” but “O” in 8-OHdG.
Line 135. what was the sensitivity of the test.
Line 146. Please precise how the OR was calculated. Was the backward procedure or another applied . For the statistical significance evaluation of the model, it could be nice to use an validation like the Wald likelihood ratio test (LR test) or something.
Results:
Line 170 and 171 please correct the symbols for proper ±.
Tables 2 and 3 not sure where are the data related to quartiles of 8-OHdG content referred in text but not in tables 2 and 3.
Table 4: provided data are inconsistent with the previous ones in Table 1. The percentages in the third column “Number of atopic phenotypes (%)” are also unknown where they arise.
Discussion:
Line 244: inflammatory process but process of inflammation
Line 253: Insert reference.
Line 267. I would recommend to change the firmness of that statement. It is formed based on the review from 2003. From that time the scientific community has shown that more and more factors make the march less obvious and some even question its legitimacy and it is postulated the coexistence of diseases of varying intensity but with lower detection in the initial phase. Also in this studies it is difficult to predict if no follow-up phase is reported.
Line 279: Some studies report a decrease in the level of this marker with age especially in the context of intestinal permeability and the role of intestinal microbiota, therefore it is worth examining the exact relationship between the direct level of factor in the group according the age.
Author Response
Open Review
Comments and Suggestions for Authors
Review: ijerph-938757
The Associations between the Levels of Urinary Oxidative Stress of 8-OHdG and the Risks of Atopic Diseases in Children
Comments and Suggestions for Authors
The paper by Pang-Yen Chen1et al. examines the role of 8-OHdG as a non-invasive marker of atopic diseases. The research questions being asked are important and experiments are pretty good executed however the originality / novelty seems average. Although the analysis of data is extensive, there are several drawbacks in the study design and presentation of the results. Statistical analysis also raises doubts.
Major Concerns and questions:
As stated above, it is not clear how some of the statistical analyses has been done.
- First: how the dominant explanatory variables for the model 1 of adjustment has been chosen. The selection should be supported with principal component analysis or other method.
Reply:
In this study, we aimed to exam the association between atopic diseases and 8-OHdG level. A conditional logistic regression would be proper for evaluating the associations. We corrected the statistical methods to make the model selection more clearly (Lines 144-159):
For Tables 2-3, the risk factors adjusted in the multivariate regression model (Table 3-Model 1) were selected from the univariate model (Table 2-Model 0) predicting any one of atopic diseases (atopic dermatitis, asthma, allergic rhinitis, night pruritus) based on the risk factor with a significance level of P-value ≤0.2 plus age and gender for all cases and controls (N=400). Model 1 (multivariate regression model) was adjusted by age, gender, daycare before 1-year-old, furry pets at home, tobacco smoke exposure during pregnancy, and maternal history of atopy (due to 8-OHdG level was correlated to AD result, the status of AD can not be considered as one of the confounder in the same model with 8-OHdG level simultaneously. As a result, the levels of continuous urinary 8-OHdG and the status of atopic dermatitis were adjusted in Model 1 independently).
Second: In several places the Tabaco smoke exposure (ETS) has been raised but it seems to be not strong association looking at case vs control values and especially that correlation coefficient is really weak r=0.14 and p=0.04.
Reply:
Thank for your comment. Since the exposures of environmental tabaco smoke were not related to the onset of AD in the logistic model. We deleted the relevant descriptions in the Discussion Section.
It has been reported that 8-OHdG and 8-OHGua is much more often noted in fluids of smokers but not in their offspring. Those results also suggested that 8-OH-Gua, rather than 8-OH-dG, may be a more general marker for oxidative damage.
Reply:
Thank you for your comment. We appreciate your suggestions; however, the study was conducted in 2011. Before 2011, few studies supported that 8-OHdG or 8-OHGua was associated with atopic diseases. In this study, we can only provide the results of 8-OHdG, and show strong association between 8-OHdG and the onset of atopic dermatitis (P<0.001). We consider the finding is worth to be reported.
Much stronger influence on atopic diseases prevalence seems to have positive history of atopy. Also on this topic, it would be more appropriate to provide the result of both parents history of atopy, not only the mothers.
Reply:
Thank you for your comment. We are sorry that we confused you in the prior version. As mentioned in the Method Section (Lines 118-119): A standardized questionnaire was used to collect information from the participants’ parents regarding the history and symptoms of atopic diseases in children. We have corrected “maternal history of atopy” to “parental history of atopy” for the whole manuscript and tables.
Third: In the comparison of some quantitative data when only case vs control has been compared the T-test should have been used, or when distribution is other than normal or variance is unequal, nonparametric Welch or Mann-Whitney tests should be applied not always Chi-square like for categorical variables.
Reply:
Thank you for your comment. We have updated the Statistical Analysis Section:
Data were expressed as mean ± standard deviation for normally distributed, continuous variables and as proportions for categorical variables. Continuous variables were analyzed using a two-tailed t-test. Non-parametric parameters were tested using the Mann-Whitney U test. Discrete variables were compared using a Chi-square test (Lines 136-139).
- Survey data and clinicians anamnesis should be also supported with some immunological factors like TEWL or declared EASI scale results in the context of reported atopic dermatitis. This would make it more credible to qualify participants to particular groups which are also not entirely clear how numerous they are.
Reply:
Thank you for your comment. A diagnosis of AD, asthma and allergic rhinitis was made by expert pediatric allergists using a standardized technique of history taking and clinical examinations. The cases of AD were confirmed according to the Hanifin and Rajka diagnostic criteria for AD. Eosinophil count, total IgE, allergen specific IgE, SCORAD score were applied in the context of reported atopic dermatitis (Lines 110-111).
It would be also good to check another
It is unclear how the number of AD people diagnosed as simultaneously suffering from asthma, allergic rhinitis night pruritus (n=49, 99, 71 respectively- Table .1) increased to 69, 115, 97 respectively (Table 4-Number of atopic phenotypes (%)).
Reply:
We are sorry for the mistake. We have corrected the number for the number of atopic phenotypes (%) in the Table 4.
- The BMI scale used in the research should be replaced with the BMI z-score coefficient recommended by WHO as more appropriate for children.
Reply:
The Z-score of BMI for children was provided in the updated Table 1. The Z-score of BMI was not associated with the onset of AD in the regression model.
- What was tested with the parameter described as the vitamin C and E intake. Was it additional supplementation with or preparations or dietary intake?
Reply:
We are sorry that we confused you in the prior version. It was daily additional supplementation. We have corrected the description in the Table 1.
In this context it would be interesting to check how the diet (that in Asian countries is reach in antioxidants) influence the usefulness of this marker.
Reply:
Thank you for your review and feedback.
- It would be also good to make it clear if the control group can be classified as a healthy because there should be no atopic cases then. That way of involving of control participants seems more clear.
Reply:
Thank you for your review and feedback.
- Were there isolated associations of marker level in relation to the age of the patients conducted or only according to the quartiles?
Reply:
Thank you for your review. In the new Table 2: Association between urinary 8-OHdG levels and the risk factors without adjustment. Age was not significantly associated with atopic diseases in the logistic regression under crude effect and adjusted effect. In this study, we discussed study findings mainly based the adjusted effect of Model 1: adjusted by age, gender, daycare <1-year-old, furry pets at home, tobacco smoke exposure during pregnancy, and parental history of atopy.
In summary, the statistical reanalyzing with better explanation of obtained data seems the most important aspect that needs to be worked on before the manuscript can be published.
Other Concerns
Introduction
Be precise that the abbreviation AD refers to atopic dermatitis and not to atopic diseases. It seems sometimes not followed by authors.
Reply:
Thank you for your review. We corrected the Introduction Section.
Line 65. Please report more up-to-date data and dedicated on the region covered by the study.
Reply:
Thank you for your review. We corrected the Introduction Section.
Materials and Methods :
Line 123: the volume of urine and amount of consumed water during the day before should be taken into consideration when the final dilution of the urine was applied.
Reply:
Thank you for your review. The concentrations of 8-OHdG were measured based the standard collection procedure and assay kit. Limitations were also provided in the prior version.
Line 124: not “0” but “O” in 8-OHdG.
Reply:
Thank you for your review. We have corrected the typo.
Line 135. what was the sensitivity of the test.
Reply:
Thank you for your comment. We have updated the Statistical Analysis Section:
Data were expressed as mean ± standard deviation for normally distributed, continuous variables and as proportions for categorical variables. Continuous variables were analyzed using a two-tailed t-test. Non-parametric parameters were tested using the Mann-Whitney U test. Discrete variables were compared using a Chi-square test (Lines 144-147).
Line 146. Please precise how the OR was calculated. Was the backward procedure or another applied . For the statistical significance evaluation of the model, it could be nice to use an validation like the Wald likelihood ratio test (LR test) or something.
Reply:
In this study, we aimed to exam the association between atopic diseases and 8-OHdG level. A conditional logistic regression would be proper for evaluating the associations. We corrected the statistical methods to make the model selection more clearly (Lines 144-159):
For Tables 2-3, the risk factors adjusted in the multivariate regression model (Table 3-Model 1) were selected from the univariate model (Table 2-Model 0) predicting any one of atopic diseases (atopic dermatitis, asthma, allergic rhinitis, night pruritus) based on the risk factor with a significance level of P-value ≤0.2 plus age and gender for all cases and controls (N=400). Model 1 (multivariate regression model) was adjusted by age, gender, daycare before 1-year-old, furry pets at home, tobacco smoke exposure during pregnancy, and maternal history of atopy (due to 8-OHdG level was correlated to AD result, the status of AD can not be considered as one of the confounder in the same model with 8-OHdG level simultaneously. As a result, the levels of continuous urinary 8-OHdG and the status of atopic dermatitis were adjusted in Model 1 independently).
Results:
Line 170 and 171 please correct the symbols for proper ±.
Reply:
Thank you for your review. We have corrected the typo.
Tables 2 and 3 not sure where are the data related to quartiles of 8-OHdG content referred in text but not in tables 2 and 3.
Reply:
Thank you for your review. For Tables 2-3, we aimed to exam the association between atopic diseases and continuous 8-OHdG level instead of quartiles for all participants (N=400) (due to 8-OHdG quartiles were different in cases and controls, it would be complex to explain the risk effect). We then tested the association between atopic diseases and 8-OHdG quartiles in Tables 4-5 for cases (N=200) and controls (N=200), independently.
The annotations of Q1-Q4 were deleted in Tables 2-3.
Table 4: provided data are inconsistent with the previous ones in Table 1. The percentages in the third column “Number of atopic phenotypes (%)” are also unknown where they arise.
Reply:
We are sorry for the mistake. We have corrected the number for the number of atopic phenotypes (%) in the Table 4.
Discussion:
Line 244: inflammatory process but process of inflammation
Reply:
Thank you for your review. We have corrected it.
Line 253: Insert reference.
Reply:
Thank you for your review. We have added the reference.
Line 267. I would recommend to change the firmness of that statement. It is formed based on the review from 2003. From that time the scientific community has shown that more and more factors make the march less obvious and some even question its legitimacy and it is postulated the coexistence of diseases of varying intensity but with lower detection in the initial phase. Also in this studies it is difficult to predict if no follow-up phase is reported.
Reply:
Thank you for your review. We have corrected it.
Line 279: Some studies report a decrease in the level of this marker with age especially in the context of intestinal permeability and the role of intestinal microbiota, therefore it is worth examining the exact relationship between the direct level of factor in the group according the age.
Reply:
Thank you for your review. We have limited case number of AD.
Firstly, stratification of age from 3-6 years may casus sporadic data, and may lead bias for statistical inference.
Secondly, age was not significantly associated with atopic diseases in the logistic regression under crude effect and adjusted effect. Besides, we discussed study findings mainly based the adjusted effect of Model 1: adjusted by age, gender, daycare <1-year-old, furry pets at home, tobacco smoke exposure during pregnancy, and parental history of atopy.
Hence, we did not examine the exact relationship between the direct level of factor in the group according to the age.
Reviewer 3 Report
This is an interesting article; however I find many limitations:
Authors should modify the conclusion in key contribution, the results of this study do not permit to treat AD examining the concentration of 8-OHdG. Nowadays treatment of AD is based in clinical findings and laboratory data do not add new information to indicate a treatment in our patients, moreover the role of IgE or eosinophils to diagnose AD is contradictory.
In the introduction, authors should mention new treatments for AD, some already approved and others in clinicals trials. The sentence “there is no treatment targeted at the basic cause of AD” is not appropriate now.
Please clarify the design of the study, a cross sectional study with a control group should be considered.
Regarding the exclusion criteria for cases, authors should have considered other inflammatory skin conditions.
Please explain properly “environment exposures” in methods.
Information regarding the severity of AD (SCORAD/EASI…) should be included. This is a very important limitation of the study. Also, information regarding treatment for AD should be included. Degree of inflammation and treatment may modify oxidate stress.
Please modify the legend of table 2 and 3 as no information appears regarding Q1-Q4, like in table 4.
Explain properly the variable “number of atopic phenotypes” in table 4.
The following sentence of the discussion is not appropriate according with study results: “urinary 8-OHdG may be considered as potential indicator in managing children with AD”
Can you explain properly why “28-OHdG via urinary samples, thereby reducing the number of unnecessary blood draws in children”. Only children under systemic treatments (cyclosporine, methotrexate, Dupilumab…) for AD requires blood draws.
Regarding the conclusion, I believe this sentence is not appropriate “the associations between the levels of urinary 8-OHdG and AD risk in children”, as this is not a prospective study analysing the risk of developing AP.
The information was obtained in 2011 and the article has been submitted 9 years later, please can you explain the delay?
Author Response
Comments and Suggestions for Authors
This is an interesting article; however I find many limitations:
Authors should modify the conclusion in key contribution, the results of this study do not permit to treat AD examining the concentration of 8-OHdG. Nowadays treatment of AD is based in clinical findings and laboratory data do not add new information to indicate a treatment in our patients, moreover the role of IgE or eosinophils to diagnose AD is contradictory.
Reply:
Thank you for your review. We revised the key contribution as:
The urinary concentration of 8-OHdG can be examined for managing AD. This study provides better understanding of the underlying mechanisms of AD and urinary 8-OHdG.
In the introduction, authors should mention new treatments for AD, some already approved and others in clinicals trials. The sentence “there is no treatment targeted at the basic cause of AD” is not appropriate now.
Reply:
Thank you for your review. We revised the Introduction Section based on your suggestions.
Please clarify the design of the study, a cross sectional study with a control group should be considered.
Reply:
Thank you for your review. As mentioned in Method Section (Lines 91-94), this is a nested case-control study (in a cohort), which can investigate a possible association between risk factor and disease. Overall 200 children were identified as AD cases. Controls were then 1 to 1 matched to the same diagnosed time of AD cases.
Regarding the exclusion criteria for cases, authors should have considered other inflammatory skin conditions.
Reply:
Thank you for your review. We revised the Method Section based on your suggestions (Line 101).
Please explain properly “environment exposures” in methods.
Reply:
Thank you for your review. We revised the Method Section (Lines, 124-125).
Information regarding the severity of AD (SCORAD/EASI…) should be included. This is a very important limitation of the study. Also, information regarding treatment for AD should be included. Degree of inflammation and treatment may modify oxidate stress.
Reply:
Thank you for your comment. We appreciate your suggestions; however, the study was conducted in 2011. In this study, we are failed to provide the severity (degree) of AD. Limitations were provided in the prior version.
Please modify the legend of table 2 and 3 as no information appears regarding Q1-Q4, like in table 4.
Reply:
Thank you for your review. For Tables 2-3, we aimed to exam the association between atopic diseases and continuous 8-OHdG level instead of quartiles for all participants (N=400) (due to 8-OHdG quartiles were different in cases and controls, it would be complex to explain the risk effect). We then tested the association between atopic diseases and 8-OHdG quartiles in Tables 4-5 for cases (N=200) and controls (N=200), independently.
The annotations of Q1-Q4 were deleted in Tables 2-3.
Explain properly the variable “number of atopic phenotypes” in table 4.
Reply:
We are sorry for the mistake. We have corrected the number for the number of atopic phenotypes (%) in the Table 4.
The following sentence of the discussion is not appropriate according with study results: “urinary 8-OHdG may be considered as potential indicator in managing children with AD”
Reply:
Thank you for your review, we have revised it as:
This study provides better understanding of the underlying mechanisms of AD and urinary 8-OHdG.
Can you explain properly why “8-OHdG via urinary samples, thereby reducing the number of unnecessary blood draws in children”. Only children under systemic treatments (cyclosporine, methotrexate, Dupilumab…) for AD requires blood draws.
Reply:
Better understanding of the underlying mechanisms of AD is important. Currently, the values of serum IgE and eosinophils in the peripheral blood are highly sensitive in detecting AD. Children under systemic treatments requires blood draws for managing AD. Considering the clinical application of urinary 8-OHdG in children with AD who need to receive blood draws during follow-up, we can simply examine the concentrations of 8-OHdG via urinary samples, thereby reducing the number of unnecessary blood draws in children (Lines 266-272).
Regarding the conclusion, I believe this sentence is not appropriate “the associations between the levels of urinary 8-OHdG and AD risk in children”, as this is not a prospective study analysing the risk of developing AD.
Reply:
Thank you for your review. This is a nested case-control study (in a cohort), which can investigate a possible association between risk factor and disease. As suggesting by other reviewers and you, I corrected the descript as “the associations between the levels of urinary 8-OHdG and the onset of AD”.
The information was obtained in 2011 and the article has been submitted 9 years later, please can you explain the delay?
Reply:
The study was conducted since I was a PhD student. I tried to publish this work after obtaining the PhD degree. However, it did not be accepted by other journals. As a result, I rewrote and try to refine this work. Hopefully, this journal can be published in this research.
Round 2
Reviewer 2 Report
I have no more comments. I think that the publication in this form may be accepted for further processing.
Author Response
Thank you for your review and considering our study for publication.
Reviewer 3 Report
Regarding the key contribution, I think that the following to sentences are inappropriate and are not related with the objective of the study: “the importance of the use of 8-OHdG as a non-invasive and convenient method to detect AD in children” and “the urinary concentration of 8-OHdG can be examined for managing AD”. More studies are necessary to confirm the use of 8-OHdG as a non-invasive method to detect AD in children and no information is included in this article regarding the use of 8-OHdG for managing AD.
Please explain in the methods new variables included: Z-score of BMI and Percentage of BMI
Patients with AD require blood draws to control possible side effects associated with systemic medication (cyclosporin, dupilumab…). I do not believe that urinary samples of 8-OHdG will reduce the number of blood draws in children with severe AD, please modify in the discussion.
Author Response
Open Review
Comments and Suggestions for Authors
Comments and Suggestions for Authors
Regarding the key contribution, I think that the following to sentences are inappropriate and are not related with the objective of the study: “the importance of the use of 8-OHdG as a non-invasive and convenient method to detect AD in children” and “the urinary concentration of 8-OHdG can be examined for managing AD”. More studies are necessary to confirm the use of 8-OHdG as a non-invasive method to detect AD in children and no information is included in this article regarding the use of 8-OHdG for managing AD.
Reply:
Thank you for your review, we softened the descriptions of the Key Contributions (Lines 51-52):
Regarding the clinical application in children with AD, the urinary concentration of 8-OHdG can be considered as one of the surrogate indexes of non-invasive method for detecting the activity of AD.
Please explain in the methods new variables included: Z-score of BMI and Percentage of BMI
Reply:
Thank you for your review, we have added in the Methods Section (Lines 130-132).
The z-scores (standard deviation scores) and percentages of BMI were measured based on the child’s BMI, age and sex, and can be checked online: https://zscore.research.chop.edu/.
Patients with AD require blood draws to control possible side effects associated with systemic medication (cyclosporin, dupilumab…). I do not believe that urinary samples of 8-OHdG will reduce the number of blood draws in children with severe AD, please modify in the discussion.
Reply:
Thank you for your review, we softened the descriptions in the Discussion Section (Lines 276-277):
Regarding the clinical application in children with AD, the urinary concentration of 8-OHdG can be considered as one of the surrogate indexes of non-invasive method for detecting the activity of AD.